# Dynamic Responsive Inguinal Scaffold Activates Myogenic Growth Factors Finalizing the Regeneration of the Herniated Groin

**DOI:** 10.3390/jfb13040253

**Published:** 2022-11-18

**Authors:** Giuseppe Amato, Giorgio Romano, Vito Rodolico, Roberto Puleio, Pietro Giorgio Calò, Giuseppe Di Buono, Luca Cicero, Giorgio Romano, Thorsten Oliver Goetze, Antonino Agrusa

**Affiliations:** 1Department of Surgical, Oncological and Oral Sciences, University of Palermo, 90127 Palermo, Italy; 2Department PROMISE, Section Pathological Anatomy, University of Palermo, 90127 Palermo, Italy; 3Department of Pathologic Anatomy and Histology, IZSS, 90129 Palermo, Italy; 4Department of Surgical Sciences, University of Cagliari, 09042 Cagliari, Italy; 5CEMERIT—IZSS, Via Gino Marinuzzi, 3, 90129 Palermo, Italy; 6Postgraduate School of General Surgery, University of Palermo, 90127 Palermo, Italy; 7Institut für Klinisch-Onkologische Forschung Krankenhaus Nordwest, 60488 Frankfurt/Main, Germany

**Keywords:** tissue degeneration, regenerative scaffolds, tissue regeneration, muscle, muscle growth factors, neo-myogenesis, inguinal protrusion disease

## Abstract

Background: Postoperative chronic pain caused by fixation and/or fibrotic incorporation of hernia meshes are the main concerns in inguinal herniorrhaphy. As inguinal hernia is a degenerative disease, logically the treatment should aim at stopping degeneration and activating regeneration. Unfortunately, in conventional prosthetic herniorrhaphy no relationship exists between pathogenesis and treatment. To overcome these incongruences, a 3D dynamic responsive multilamellar scaffold has been developed for fixation-free inguinal hernia repair. Made of polypropylene like conventional flat meshes, the dynamic behavior of the scaffold allows for the regeneration of all typical inguinal components: connective tissue, vessels, nerves, and myocytes. This investigation aims to demonstrate that, moving in tune with the groin, the 3D scaffold attracts myogenic growth factors activating the development of mature myocytes and, thus, re-establishing the herniated inguinal barrier. Methods: Biopsy samples excised from the 3D scaffold at different postoperative stages were stained with H&E and Azan–Mallory; immunohistochemistry for NGF and NGFR p75 was performed to verify the degree of involvement of muscular growth factors in the neomyogenesis. Results: Histological evidence of progressive muscle development and immunohistochemical proof of NFG and NFGRp75 contribution in neomyogenesis within the 3D scaffold was documented and statistically validated. Conclusion: The investigation appears to confirm that a 3D polypropylene scaffold designed to confer dynamic responsivity, unlike the fibrotic scar plate of static meshes, attracts myogenic growth factors turning the biological response into tissue regeneration. Newly developed muscles allow the scaffold to restore the integrity of the inguinal barrier.

## 1. Introduction

The principle of deploying a flat mesh made of biocompatible material to reinforce the groin affected by hernial protrusion was proposed in the 1960s in contraposition to the so-called pure tissue repair [1,2]. Over the years, various synthetic materials have been proposed for manufacturing prostheses in hernia surgery [3]. In the past century, polymers such as nylon, dacron, and polyester were used as hernia meshes but, thanks to enhanced biological performances after implantation, the most popular material employed for manufacturing prosthetic devices is polypropylene, which still continues to be widely used for this purpose [4]. Prosthetic repair of inguinal hernias has proven effective in significantly reducing the rate of recurrence and, thus, has progressively gained popularity among surgeons. Nowadays, prosthetic open or laparoscopic repair with flat meshes is almost unanimously considered the treatment of choice for the cure of groin hernias [5,6]. Nevertheless, despite wide popularity, increasing concerns negatively influence this model of therapy. In particular, mesh fixation and low-quality biologic response with uncontrolled fibrotic ingrowth, postoperative discomfort and chronic pain have increasingly fallen under the spotlight of researchers [7,8,9]. To date, no effective solutions have been proposed to resolve these controversial outcomes. Recently, following new studies on the physiopathology and functional anatomy of the herniated groin, a different line of thought has emerged [10,11,12,13,14,15]. The results of these investigations have led to a different concept for the therapy of inguinal protrusions. According to these studies, a novel device was developed for inguinal herniorrhaphy [16]. Named ProFlor, it is made of the same polypropylene material as conventional flat meshes but has a 3D multilamellar design with a compressible outline conferring dynamic responsivity. Thanks to this proprietary outline it works on a completely different principle. Actually, once positioned into the hernia opening, the inherent centrifugal force expands the 3D device to permanently obliterate the defect in fixation-free fashion. Its dynamic responsivity consents a fully different biological response, allowing the incorporation into the 3D device of well-hydrated connective tissue as well as new arteries, veins, nerves, and muscles, thus re-establishing the integrity of the herniated inguinal barrier [17,18,19]. Additional evidence of the dynamically induced regenerative features of ProFlor in patients has been confirmed by signal intensity assessment in MR imaging [20]. To further explain the regenerative features, a recent investigation has documented the involvement of vascular growth factors in modulating the probiotic response of these tissue components [21]. However, despite there being a solid neomyogenesis which has already been documented in ProFlor, knowledge is still lacking concerning the myogenic factors involved in neomyogenesis inside the 3D dynamic responsive hernia scaffold [22]. This investigation aims to demonstrate the presence of myogenic active growth factors, such as nerve growth factor (NGF) and the neurotrophin receptor NGFR p75, that along all stages of the postoperative course actively contribute to modulating and finalizing muscle regeneration within ProFlor. 

## 2. Material and Methods

This multicentric study was approved by the Ethics Commission of the Medical Board of the Land Hessen, Germany (Approval Number: FF32/2013). The research and related investigative methods were registered on ClinicalTrials.gov (https://clinicaltrials.gov/ct2/show/NCT05072171?term=NCT05072171&draw=2&rank=1), identifying number ID: NCT05072171—accessed on 8 October 2021. 

The research was carried out in accordance with the Declaration of Helsinki for experiments involving humans. Written informed consent was obtained from each participant involved in the study. 

The investigation focused on individuals all diagnosed with primary uncomplicated inguinal protrusion who underwent hernia repair with the 3D dynamic responsive hernia scaffold ProFlor. (Figure 1A) The study utilizes the same tissue specimens used for other studies and excised from fifteen patients who underwent primary inguinal hernia repair in the period 2010–2015 and that, for different reasons, needed additional surgery for different reasons in the previously operated groin [17,18,19,20,21,22]. The mean age of the patients was 52 years, mean BMI 27.6. Nine hernia defects were dimensioned between 31 and 37 mm while the remaining six between 25 and 30 mm. ProFlor is composed of a multilamellar cylindrical 3D core manufactured with low-weight, large porous polypropylene. The lamellas have a thickness of 15 mm and reinforced edges, obtained by rolling and welding with ultrasound the external margins. The lamellas are connected at the center to two rings of polypropylene in the shape of petals to form a flower-like 3D core. This central core has two dimensions: 25 mm (made up of 6 petals) and 40 mm (made up of 8 petals). The 3D core of ProFlor is designed to be compressible on both planes, longitudinal and transversal. One surface of the 3D scaffolds is connected at the center to a flat mesh of different sizes. The 25 mm unit has a 60 mm rounded flat portion, the rounded flat mesh of the 40 mm sized scaffold measures 70 mm, while the ProFlor E (extended) version usually used for large hernia defects and for laparoscopic procedures has an oval shape measuring 80 × 100 mm. The connected flat mesh is positioned to prophylactic cover the posterior inguinal backwall facing the peritoneum. Thanks to its inherent centrifugal expansion, it is delivered fixation free, to permanently obliterate the hernial opening (Figure 1B). Once delivered, ProFlor moves compliant to the groin’s movements.

The tissue specimens were excised at defined stages post implantation three in the short term (ST) between three and five weeks, five in the mid-term between three and four months (MT) post implantation, four in the long-term post implantation, (LT) between six and eight months, and the final three in the extra-long term (ET) post implantation, more than three years after delivery. The tissue specimens were excised from the frontal aspect of ProFlor, which corresponds to the surface of the device that faces the fascia of the external oblique muscle.

### 2.1. Histological Evaluation 

Once excised, the biopsy samples were first secured in neutral-buffered 10% formalin then embedded in paraffin wax (Figure 1C). Sections (4 μm thick) were prepared, kept at room temperature, and stained with hematoxylin and eosin (H&E) and Azan–Mallory trichrome (Bio-Optica Milano S.p.A. Via San Faustino 58—20134 Milano—Italy).

### 2.2. Immunohistochemistry

Additional sections were processed to assess at each temporal stage:(a)The presence of neomyogenetic activity by evidencing the density of specific clusters using an anti-NGF rabbit monoclonal antibody (Abcam, 52 Grove Street Waltham, MA 02453. USA—code: ab52918), dilution: 1:100;(b)Cross evidence of the NGF active clusters by detecting the NGF receptors in the excised tissue with NGFR p75 mouse monoclonal antibody (Santa Cruz Biotechnology, Inc. 10410 Finnell Street Dallas, Texas 75220 U.S.A., code: sc.13577), dilution: 1:100.

The slides were then dewaxed, and heated for antigen retrieval in a Tris EDTA solution (pH 6.0) at 96 °C for 20 min. A solution of 3% *w*/*v* hydrogen peroxide diluted in methanol served for blocking endogenous peroxidase activity. Slides were treated with a background sniper for fifteen minutes for reducing nonspecific background staining. Afterwards, incubation with primary antibodies (NGF, NGFR p75) for r 60 min at room temperature followed. At this stage, the slides were rinsed three times for five minutes with phosphate-buffered saline (PBS) f, and a secondary biotinylated immunoglobulin (LSAB, Dako—Agilent5301 Stevens Creek Blvd. Santa Clara, CA 95051, USA) was applied for thirty minutes at RT. Tissue sections were rinsed for five minutes in PBS, then incubated in PBS at room temperature with streptavidin and horseradish peroxidase conjugate for 60 min. All tissue sections were rinsed three times with Tris-buffer saline (TBS), incubated with the chromogen 3-3′-diaminobenzidine tetrahydrochloride (DAB, Dako—Agilent 5301 Stevens Creek Blvd. Santa Clara, CA 95051 USA) diluted 0.035% in TBS for 1 min, rinsed in tap water, and counterstained with Mayer’s hematoxylin. When positive, the DAB-reaction showed a brown precipitate. The specific primary antibodies were replaced with PBS or normal goat serum in tissue sections used as negative controls [23]. A Leica DMR microscope equipped with a Nikon DS-Fi1 digital camera was used to analyze at a 200× magnification all immune-stained samples.

### 2.3. NGF and NGFR Quantification

Tissue sections stained with anti-NGF and NGFR p75 antibody were observed at 200× magnification and three non-overlapping fields were randomly evidenced. Microphotographs were then processed with Image J computer application to assess the rate of immunohistochemical positive cells to NGF and NGFR p7.

### 2.4. Statistical Analysis

NGF and NGFR p75 immunohistochemical positive cells evidenced in 15 tissue specimens were excised from the ProFlor at different postoperative stages. Three units were removed three to five weeks postoperative (ST), four units three to four months postoperative (MT), five samples six to eight months LT post-implantation and three ET post-implantation, more than three years after implantation, were analyzed. To detect Gaussian distribution of data, a normality test (Shapiro–Wilk test) was performed, which resulted positive. To compare NGF and NGFR p75 immunohistochemistry between 15 tissue specimens of ProFlor along the periods (ST, MT, LT, ET postoperative), a one-way ANOVA test was conducted. Furthermore, to assess the significance of differences between the defined periods the Tukey–Kramer multiple comparison test was used. A *p* value < 0.05 was considered worthy of note. A Pearson correlation coefficient was computed to assess the relationship between NGF and NGFR p75 to evaluate the strength of the linear relationship between the two variables. GraphPad Prism 8.1.1 Software (GraphPad, San Diego, CA, USA) was used for the analyses.

## 3. Results

### 3.1. Histological Assessment

In the histopathological evaluation of the H&E-stained slides, specimens excised at 3–5 weeks ST post-implantation showed mild lymphohistiocytic inflammatory infiltrate and a proliferation of fibroblasts in a context of connective tissue incorporation. Angiogenetic clusters served to support the newly ingrowing tissue. (Figure 2A) The H&E-stained slides of the specimens, resected 3–4 months MT post-implantation, evidenced an increased number of neoangiogenetic nuclei in a frame of collagen synthesis and connective tissue remodeling. Fibroblast proliferation and newly formed myocytes were also detectable within the polypropylene structure of ProFlor. (Figure 2B) Six to eight months LT post-implantation, within the polypropylene structure of the 3D scaffold, the presence of well-structured muscle bundles supported by mature vascular and nerve elements could be detected (Figure 2C). More than three years after implantation (ET) fully developed multinucleated myocytes confirmed the ultimate regeneration of newly formed muscle elements, as well as well-structured nerves and vessels close to the polypropylene fibers of the 3D scaffold (Figure 2D).

In particular, regarding neomyogenesis, H&E staining revealed the progressive development of mature cells from stem cells having, first, the appearance of myoblasts that in MT and LT completed differentiation into fully functional myocytes. In the ET, these already differentiated myoblasts assumed the aspect of multinucleated myofibers. Therefore, all stages of a fully accomplished muscle regeneration could be seen. It was also interesting to note the presence of well-structured nerves starting from the MT as well as circumscribed areas of regenerated mature adipose tissue in the ET samples.

These histological finding were confirmed by analysis of the Azan–Mallory trichrome staining that showed the progressive ingrowth of muscular elements in a surround of contextual development of connective and vascular elements. Specifically, starting from the proliferation of fibroblasts and the initial myocytic differentiation of the early stage, all steps of muscle development could be followed up along all temporal stages under scrutiny (Figure 3A–D).

### 3.2. Immunohistochemistry

Examination of the NGF-stained specimens excised in the ST post-implantation showed several positive immunostained cells in the area close to the polypropylene fabric of ProFlor (Figure 4A). NGF positivity of the excised specimens showed to be progressively and unceasingly increasing in the MT and ET (Figure 4B–D).

Concerning NGFR p75 immunostaining, unlike NGF, this element was scarcely evidenced in the initial stages of the ST and MT (Figure 5A,B). On the contrary, NGFR p75 was clearly noticeable from the LT, and progressively increased to the ET where a significant presence of NGFR p75 positive cells could, unmistakably, be noticed (Figure 5C,D).

Overall, mesenchymal spindle cells close to the scaffold were consistently positive for NGF and NGFR p75; in particular, immunoreactivity to these growth factors could be observed on the cell membrane and in the cytoplasm in LT and ET.

### 3.3. Statistical Analysis

The average percentage immunohistochemical positive cells to NGF and NGFR p75 of 15 biopsies excised from the 3D dynamic hernia scaffold at different stages post implantation were compared with a one-way Anova and post hoc Tuckey test.

Results showed statistically significant differences between stages and, in particular, suggest that both NGF and NGFR p75 seen in the 3D dynamic scaffold for inguinal repair increased over time (Figure 6A,B). Specifically, our results highlight that NGF is significantly greater in the LT than in the MT (*p* < 0.0001), while NGFR p75 rose significantly more in the ET than in the LT. However, there is no statistical difference in NGF-assessed LT versus ET, and in NGFR p75 ST versus MT. A Pearson correlation coefficient was computed to assess the relationship between NGF and NGFR p75. There was a positive correlation between the two variables, r = 0.786, *p* < 0.0001. A scatterplot summarizes the results (Figure 7). Overall, there was a strong, positive correlation between NGF and NGFR p75, in particular, beyond the LT stage. Increases of NGF were correlated with contextually increasing NGFR p75.

## 4. Discussion

Prosthetic repair has been acknowledged as the standard cure for the treatment of inguinal hernia for decades. Anterior or posterior (open or laparoscopic) deployment of flat static meshes to cover the inguinal area constitutes the essence of the procedure. The principle of this treatment is strengthening the weakened groin through the fibrotic tissue enfolding of the mesh. However, sometimes reactive fibrosis is excessive and may incorporate or compress the highly sensitive nerves of the groin. This is unanimously considered the culprit of the life-wasting post-herniorrhaphy chronic pain syndrome that occurs in up to 20% of herniorrhaphies [7,8]. The mesh material used is almost exclusively polypropylene, although other materials—resorbable or not—have been proposed over the years [24]. From a physical standpoint, conventional hernia meshes are thin, flat, and passive. Being static, these implants do not participate in the natural movements of the groin, which is one of the most motile parts of the body [16]. The negative fallouts of this physical incongruence are worsened by the need for mesh fixation, which increases perioperative pain and complication rates [25]. Apart from these specific mesh-related complications, it would appear obvious that the concept of conventional prosthetic hernia repair is unconnected with the degenerative pathogenesis of inguinal hernia disease [26]. Logically, the cure of a degenerative disease that produces a defect in the inguinal barrier should aim to permanently fill the hole and regenerate the wasted muscular barrier. In this regard, it should be stressed that stiff fibrotic scar ingrowing in flat static meshes merely corresponds to a common foreign body reaction, not to tissue regeneration. With this in mind, a new concept based on an updated physiological and pathogenetic model has been conceived for repairing groin hernias. A 3D multilamellar scaffold made of the same material as conventional flat hernia meshes, polypropylene, with an intrinsic compressibility has been developed for hernia repair. This device, named ProFlor, is delivered fixation free with an open and laparoscopic approach to permanently obliterate the hernia defect thanks to an inherent centrifugal expansion [27,28,29]. It is designed to be dynamically responsive since it compresses and relaxes in synchrony with the groin. ProFlor dynamic responsivity makes the difference in terms of biologic response. While manufactured with the same polypropylene material as conventional meshes, what effectively develops in the 3D hernia device is not the typical fibrotic scar plate of a foreign body reaction, but quite a different tissue [18]. From early-stage post-implantation, arteries, veins, connective tissue, nerves, and muscles progressively develop until full maturation within the ProFlor device. This corresponds to the portrait of a regenerative process that ultimately re-establishes the integrity of the myotendineal barrier of the groin [19]. Again, the sole difference between the two types of hernia devices is the physical aspect: conventional flat meshes are static and passive, while ProFlor is dynamically responsive and moves together with the groin. Nevertheless, to investigate the details of the uncommon biological response of ProFlor when delivered into the highly motile area of the groin, proof of significant steps of tissue regeneration beyond the already proven neo-angiogenesis, neo-nervegenesis, and neo-myogenesis was deemed necessary [20,21,22]. One attempt to reveal if growth factors contribute to the regenerative effect mediated by the dynamic responsivity of the 3D scaffold has been successfully carried out. It concerned the undisputable involvement of vascular growth factors such as VEFG, CD31, and SMA involved in the development of a well-structured vascular system inside the newly ingrown tissue in the ProFlor device [21].

However, to fully validate ProFlor as a regenerative hernia scaffold, a natural extension of the previous scientific experiences involved evidencing myogenic growth factors in activating development until full maturation of newly formed myocytes, the main elements of the inguinal barrier.

Muscle regeneration occurs through proper activation, development, and characterization of myogenic precursors, whose regenerative attitude is specifically managed by cytokines and growth factors contained in the myogenic precursor contained in the myocytic microenvironment [30]. During myocytic development, muscle fibrils activate a series of neurotrophins, among these NGF, and their receptors, also involving tyrosinekinase receptors (TrkA and TrkB) and p75-neurotrophin receptor (p75NTR) [31,32]. Several studies have demonstrated that cellular signaling pathways are activated by the neurotrophin/NGFR p75, which “*in vitro*” promotes myogenic differentiation, myotube survival, and “*in vivo*” repair of damaged muscle fibers [31,32,33].

NGF is a neurotrophin well known for its unquestionable role in modulating development, reliability and survival of nervous elements [34]. However, many studies confirm that NGF is also involved in many significant functions in many types of tissue structures. NGF signaling is activated by two types of receptors: the Trk receptor (tropomyosin-related kinase) and the NGFR p75 (p75 neurotrophin receptor) [35].

It has now been well established that the precursor form of NGF (proNGF) binds and activates NGFR p75. The mechanism activated by NGFp75 may differ depending on the specific cellular environment, the co-existence of activating proteins, the contextual presence of Trk receptors, and the quantitative relationship between the already developed neurotrophin and its specific originating elements [36].

NGF seems to be particularly involved in muscle regeneration; it has been demonstrated that anti-NGF transgenic mice develop atrophy of the muscle bundle, embodied by the presence of inflammatory clusters and fibrotic degeneration resembling a dystrophic phenotype [37]. NGF is predominantly evident in the developing myocytes. This occurrence strengthens the hypothesis that neurotrophin may activate and implement regeneration of the muscular elements [38]. Plenty of experimental evidence prove that NGF induces probiotic features on myofibrils by binding to NGFR p75, especially if absent TrkA or very low evidence of this protein has been detected [33].

Notably, another study demonstrated that the proNGF/NGFR p75 mode of action could have a noteworthy impact in the physiology of muscle cells, as specific activation of NGFR p75 signal is essential for an adequate structural characterization of the muscular elements [39].

NGF played a positive role in myogenesis by inducing a significant rise in the amount of myoblasts containing more than 20 nuclei. The hypertrophic effect of the NGF/p75NTR axis due to both an improved fusion rate and increased amount of fibers was further confirmed in an *in vitro* experiment [35]. Moreover, NGFR p75 is considered a marker for regenerating fibers in inflammatory muscle dystrophy [40]. Experimental investigations demonstrate that NGF appears to activate muscle regeneration and may increase the regenerative attitude of muscle stem cells in dystrophic muscle diseases [37].

With reference to the results of the present investigation, it has become evident that the progressive development until full maturation of muscular elements, clearly shown by H&E and AM microphotographs provided herewith, should be supported by myogenic growth factors. Evidence within the implanted ProFlor device of NGF positive areas that, over time, progressively increase, seems to confirm this hypothesis. NGF immunochemistry on ProFlor specimens, followed up from the ST, through MT, LT, and ET post implantation, demonstrated the progressive imprinting of this growth factor in newly developed tissue in the 3D hernia device. Further confirmation of NGF mediated muscle development derives from the immunohistochemistry of ProFlor biopsies at correspondent postoperative stages carried out with NGFR p75. These microphotographs showed very few positive areas in the early and MT stages, while, in the MT, the areas progressively increase until they reach the widespread positivity highlighted in the ET. This event may have a specific explanation: the literature suggests that NGFR p75 is the mediator of nerve–muscle communication, which can be activated only when myocytes effectively achieve functionality [41]. The lack or scarce presence of NGFR p75 during the early phases post-implantation indirectly confirms that the myogenetic process needs at least 6 months until myocytes are developed enough to receive nerve impulses and, thus, interact with the nervous network. 

This investigation shows a major limitation that concerns the apparently limited cohort of patients undergoing biopsy. Nevertheless, the collection of biopsy samples from subjects who, for different reasons underwent, a surgical revision of the groin previously operated with ProFlor was not a daily occurrence. Actually, it took several years to reach an adequate number of specimens. We are confident, however, that the 15 biopsy samples analyzed ultimately represent a statistically significant amount of data.

## 5. Conclusions

All of the above would appear to confirm that, unlike the ordinary 50-year-old treatment concept with conventional flat static meshes, the 3D dynamic responsive structure of the ProFlor device attracts tissue growth factors acting as a true regenerative scaffold. The updated physiological and pathogenetically coherent treatment concept embodied by ProFlor seems to represent a game changer in the intricate realm of inguinal hernia repair. Effectively, ProFlor probably embodies a new category of hernia devices presenting with completely new and updated concepts for the treatment of inguinal protrusions: fixation-free deployment, permanent defect obliteration, dynamic responsive behavior, and regenerative scaffold. These innovative concepts constitute a turning point for the cure of inguinal hernia and may represent the basis for the development of more advanced devices for the future therapy of this frequently occurring disease.

## Figures and Tables

**Figure 1 jfb-13-00253-f001:**
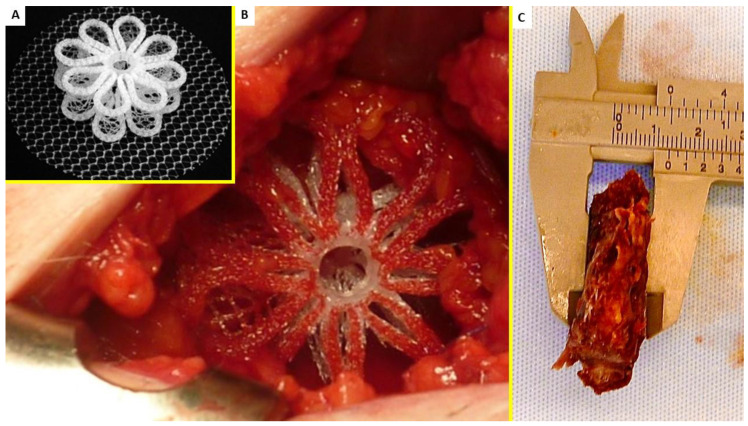
(**A**) The outline of ProFlor, composed of a multilamellar cylindrical 3D core, 15 mm thick, available in 2 different sizes: 40 and 25 mm. Edges of the lamellae are reinforced to confer springiness. The posterior surface is connected with a flat disc intended to face the peritoneal sheath. The 3D scaffold and preperitoneal disc are made from low-weight, large porous polypropylene. (**B**) The 3D ProFlor hernia scaffold just positioned to obliterate a direct hernia defect in the right groin. (**C**) The 3D hernia scaffold ProFlor removed following recurrence 3 years after implantation. ProFlor was already immersed in formalin for the histological study. Fabric of the device is fully incorporated by newly ingrown fleshy tissue and is no longer recognizable.

**Figure 2 jfb-13-00253-f002:**
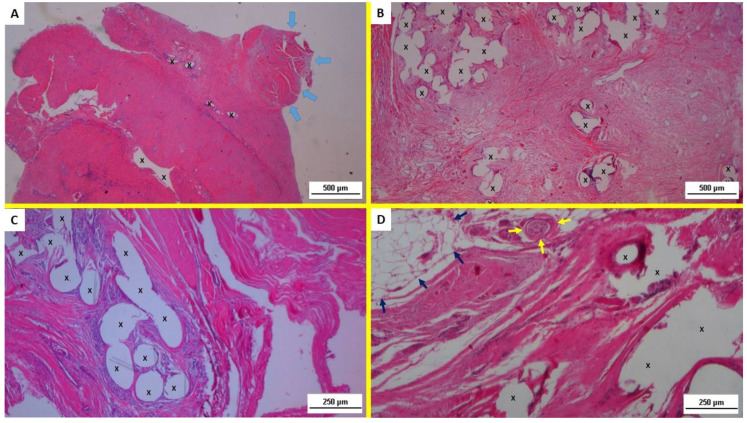
(**A**) Biopsy specimen removed from ProFlor 3 weeks postop. (ST): microphotograph shows a proliferation of fibroblasts and a mild inflammatory reaction within the polypropylene fabric (X). Muscle bundles in stage of development are present at the top right of the section (blue arrows). HE 25×. (**B**) Biopsy specimen resected from ProFlor 3 months postop. (MT): evident proliferation of fibroblasts and newly formed myocytes within the polypropylene structure of ProFlor (X). Several nuclei of neoangiogenetic clusters, collagen synthesis, and connective tissue remodeling are also present. He 25×. (**C**) Biopsy specimen excised from ProFlor 7 months postop. (LT): manifest, plentiful presence of mature muscle bundles (colored in red) around polypropylene structure of 3D scaffold (X). HE 50×. (**D**) Resected specimen from ProFlor 38 months postop. (ET): consolidated muscle regeneration represented by fully developed multinucleated myocytes close to polypropylene fibers of scaffold (X); a regenerated nerve axon enfolded by myelin sheath is also present (yellow arrows). In the contiguous area at the top left of section, mature adipose tissue is also represented (blue arrows). He 50×.

**Figure 3 jfb-13-00253-f003:**
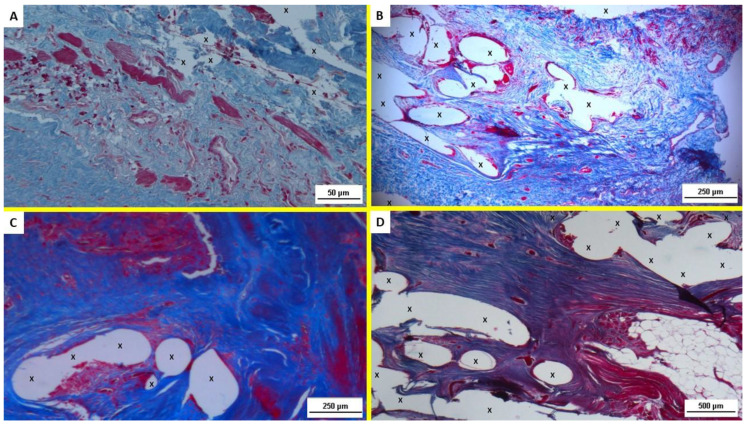
(**A**) Biopsy specimen excised from ProFlor 4 weeks postop. (ST): proliferation of fibroblasts (blue color) and ingrowing muscle fibers (colored in red) close to polypropylene fabric (X), in a context of woven, well-perfused connective tissue. AM 200×. (**B**) Biopsy specimen excised from ProFlor 4 months postop. (MT): combination of neoangiogenesis, fibroblast proliferation, collagen synthesis, tissue remodeling leading to connective tissue ingrowth (blue colored areas), and neomyogenesis (red colored structures) among polypropylene fabric of 3D scaffold (X). AM 50×. (**C**) Biopsy specimen resected from ProFlor 8 months postop. (LT): abundance of fully developed muscle fibers (red color) around polypropylene fibers of 3D scaffold (X). AM 50×. (**D**) Biopsy specimen resected from ProFlor 38 months postop. (ET): muscle regeneration represented by mature multinucleated myocytes (colored in red) developed inside polypropylene fibers of 3D scaffold (X); fibroblast proliferation and connective tissue remodeling (colored in blue) is evident. Circumscribed area of regenerated mature adipose tissue is also present at the bottom right of section. AM 25×.

**Figure 4 jfb-13-00253-f004:**
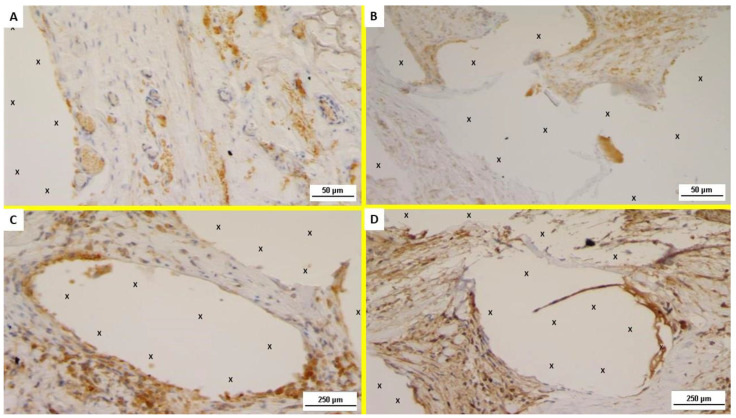
Immunohistochemistry (IHC) for NGF. (**A**) Biopsy specimen excised from ProFlor 3 weeks postop. (ST): increasing number of positive cells for NGF close to polypropylene fabric (X). NGF 200×. (**B**) Biopsy sample excised from ProFlor 3 months postop. (MT): increasing number of positive cells for NGF close to polypropylene fabric (X). NGF 200× (**C**) Biopsy specimen excised from ProFlor 7 months postop. (LT): Several NGF positive cells close to polypropylene fibers (X). NGF 50×. (**D**) Biopsy sample excised from ProFlor 38 months postop. (ET): noteworthy evidence of NGF positive cells close to polypropylene fabric (X). NGF 50×.

**Figure 5 jfb-13-00253-f005:**
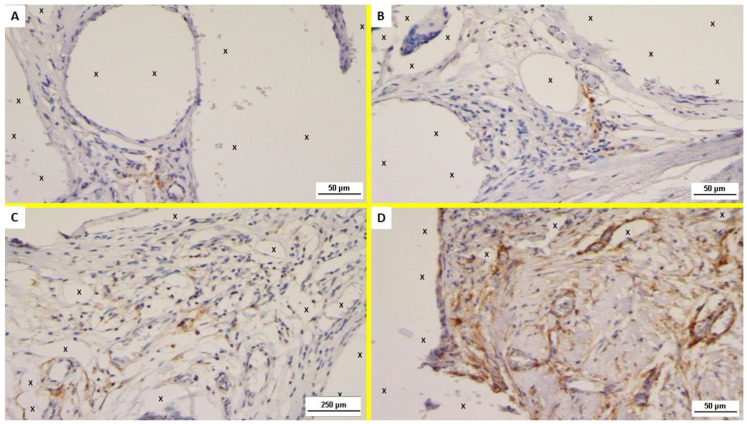
Immunohistochemistry (IHC) for NGFR p75 (**A**) Biopsy specimen removed from ProFlor 4 weeks postop. (ST): very low number of positive cells for NGFR p75 in the area close to the 3D polypropylene scaffold (X). NGFR p75 200×. (**B**) Biopsy sample excised from ProFlor 4 months postop. (MT): limited number of NGFR p75 positive cells in the context of ProFlor (X). NGFR p75 200×. (**C**) Biopsy specimen excised from ProFlor 8 months postop. (LT): Long-term microphotograph (20×). A clearly increased number of NGFR p75 positive cells close to 3D scaffold structure (X). NGFR p75 50×. (**D**) Biopsy sample resected from ProFlor 42 months postop. (ET): microphotograph shows significant presence of NGFR p75 positive cells close to polypropylene fibers of 3D scaffold (X). NGFR p75 200×.

**Figure 6 jfb-13-00253-f006:**
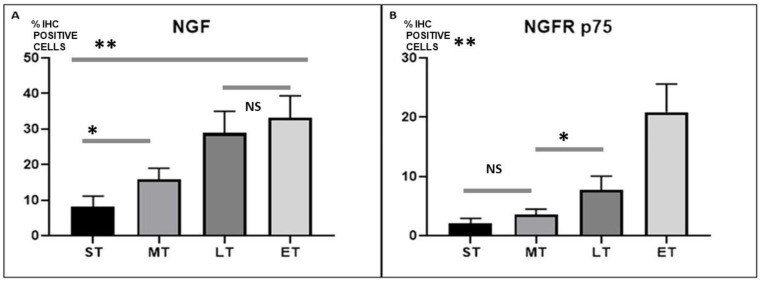
(**A**) Average percentage (SD) of immunohistochemical positive cells to NGF in the ST, MT, LT, and ET. Regular increase in NGF positive cells from first step (ST) to last step (ET) (* = *p* < 0.05, ** = *p* < 0.001, NS = no statistically relevant difference). (**B**) Average percentage (SD) of immunohistochemical positive cells to NGFR p75 in the ST, MT, LT, and ET. In the first three stages, a small increase in immunohistochemical positive cells is observed, while in the ET period the increase is much more evident. (* = *p* < 0.05, ** = *p* < 0.001, NS = no statistically relevant difference).

**Figure 7 jfb-13-00253-f007:**
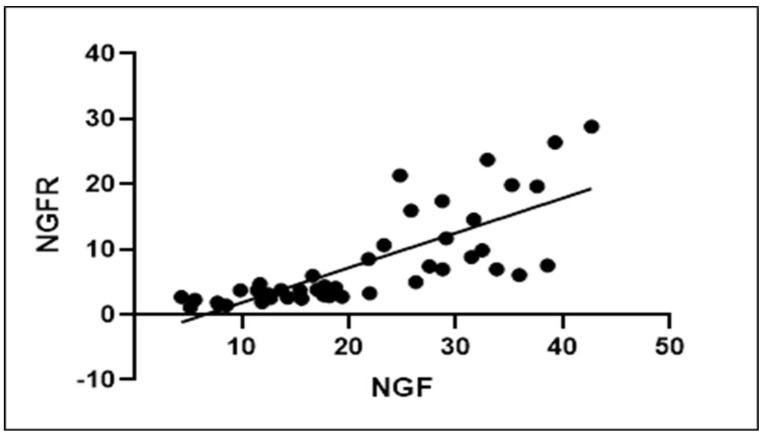
Pearson correlation coefficient between NGF and NGFR p75. A positive correlation between the two variables (r = 0.786, *p* < 0.0001) is clearly evident.

## Data Availability

The data presented in this study are available on request from the corresponding author.

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
