# Peer review of "Dynamic Responsive Inguinal Scaffold Activates Myogenic Growth Factors Finalizing the Regeneration of the Herniated Groin"

_jfb, 2022, doi:10.3390/jfb13040253_

Round 1
Reviewer 1 Report
The manuscript entitled “Dynamic responsive inguinal scaffold activates myogenic growth factors finalizing the regeneration of the herniated groin” described the development of 3D scaffold that can aid in regeneration of connective, vessels, nerves and myocytes in the treatment of inguinal hernia. However, the manuscript suffers from a number of shortcomings as described below:
1. The authors are requested to change the line 37 and 38 for better and easy understanding of the readers. [Page 2]
2. The authors should also tell if it is just “connective” or “connective tissue”?
[Line 39 -Page 2]
3. What are the dimensions of the flat mesh? [Line 108 -Page 4]
4. On what basis the longitudinal dimensions 25 or 40 mm is chosen for clinical trials [Line 108-Page 4]
5. The purpose of background sniper can be added for the readers to understand.
[Line 137-Page 5]
Author Response
Academic Editor Comments
Some comments to improve the quality of the manuscript in order to better fit the scope of the special issue.
- More information should be reported about the scaffolds fabrication/characterization;
R: the part of Material and methods dealing with the detailed description of the structure of ProFlor has been modified to fulfill this requirement
- The graphical abstract should be revised in a more concise and synthetic form;
R: Done
- The abstract should be revised in the form requested by the journal.
R: Done
Reply to Reviewer #1
Comments and Suggestions for Authors
The manuscript entitled “Dynamic responsive inguinal scaffold activates myogenic growth factors finalizing the regeneration of the herniated groin” described the development of 3D scaffold that can aid in regeneration of connective, vessels, nerves and myocytes in the treatment of inguinal hernia. However, the manuscript suffers from a number of shortcomings as described below:
- The authors are requested to change the line 37 and 38 for better and easy understanding of the readers. [Page 2]
R: Lines 37 and 39 have been amended following the suggestion of the reviewer
- The authors should also tell if it is just “connective” or “connective tissue”?
[Line 39 -Page 2]
R: In the text is now specified connective “tissue”
- What are the dimensions of the flat mesh? [Line 108 -Page 4]
R: There are different dimensions of the flat part of the device. The 25 mm unit has a 60 mm rounded flat portion, the flat mesh of the 40 mm sized scaffold measures 70 mm, while the ProFlor E (extended) version usually used for large hernia defects and for laparoscopic procedures has an oval shape measuring 80 x 100 mm. This part of the manuscript has been modified accordingly.
- On what basis the longitudinal dimensions 25 or 40 mm is chosen for clinical trials [Line 108-Page 4]
R: the scaffold type is chosen depending on the dimension of the defect. Defects smaller than 23 mm are managed with the 25 mm type. Defects larger than 24 mm are repaired with the 40 mm type.
- The purpose of background sniper can be added for the readers to understand.
[Line 137-Page 5]
R: In the text has been specified that the slides were treated with background sniper for 15 minutes
Reviewer 2 Report
Manuscript “Dynamic responsive inguinal scaffold activates myogenic growth factors finalizing the regeneration of the herniated groin” offers an interesting insight into ProFlor device and its properties and application. It is well conceptualized study with interesting results and prospects. I truly enjoyed reading it and I strongly recommend it for publishing. However, I do have couple of comments/suggestions:
11. Introduction is very well written, it covers everything relevant for the topic and it is not too long nor hard to follow. There are other materials used for these purposes so far (woven nylon, polyester, animal derived tissue)? It might be useful to add that in Introduction together with the explanation why the option you have chosen works the best.
22. L229 – 234 references needed.
33. Are there any previous publication on ProFlor device? If yes, please refer to them in the discussion section (specifically, L242 – 250).
44. Make sure that you use italic for Latin terms such as in vivo/in vitro etc.
Author Response
Comments and Suggestions for Authors
Manuscript “Dynamic responsive inguinal scaffold activates myogenic growth factors finalizing the regeneration of the herniated groin” offers an interesting insight into ProFlor device and its properties and application. It is well conceptualized study with interesting results and prospects. I truly enjoyed reading it and I strongly recommend it for publishing.
R.: we feel really thankful to the reviewer for the positive evaluation of our work.
However, I do have couple of comments/suggestions:
- Introduction is very well written, it covers everything relevant for the topic and it is not too long nor hard to follow. There are other materials used for these purposes so far (woven nylon, polyester, animal derived tissue)? It might be useful to add that in Introduction together with the explanation why the option you have chosen works the best.
R.: The section Introduction has been modified by adding a couple of sentences dealing with this matter. Related references also justify the improvements.
- L229 – 234 references needed.
R.: done, three specific references have been indicated to confute the statements made. Reference list has been modified accordingly
- Are there any previous publication on ProFlor device? If yes, please refer to them in the discussion section (specifically, L242 – 250).
R.: In reference list the articles now quoted from 16 to 22 deal with clinical and biological features of ProFlor. However, three more articles on this topic have been added and quoted in the reference list
- Make sure that you use italic for Latin terms such as in vivo/in vitro etc.
R.: The terms “in vitro” and “in vivo” at line 270 have been changed in italic font as requested.
Reviewer 3 Report
A brief summary
The manuscript entitled ‘Dynamic responsive inguinal scaffold activates myogenic growth factors finalizing the regeneration of the herniated groin’ demonstrates results of inguinal hernia repair using 3D polypropylene scaffold. However, in my opinion before publication a more detailed description of experimental procedure should be provided.
General concept comments
The section “Material and Methods” should be enhanced information about patients’ cohort. It seems to me, the next data is needed to accurate interpretation of results: patients’ age, size of inguinal hernia and strangulated or not hernia.
The authors use parametric tests such as ANOVA and Tukey for statistical analysis. It is known that these criteria are sensitive to the type of data distribution, while the authors do not assess the normality of the distribution. According to the Materials and Methods section, the statistical analysis is carried out on 9 images for two groups (ST and ET). For such samples, an analysis of the normality of the distribution should be carried out.
It is not clear, what is the mechanism of increasing the nerve growth factor and the neurotrophin receptor? Was there a control group of patients who were implanted with a standard mesh? How is level of the nerve growth factor for these patients? Please, discuss this point.
Specific comments
Figures 2, 3, 4 and 5: Images on these panels have various magnification. Please unify.
Figure 6: Please add name of Y-axes on bar graphs.
Author Response
Reply to reviewer #3
Comments and Suggestions for Authors
The manuscript entitled ‘Dynamic responsive inguinal scaffold activates myogenic growth factors finalizing the regeneration of the herniated groin’ demonstrates results of inguinal hernia repair using 3D polypropylene scaffold. However, in my opinion before publication a more detailed description of experimental procedure should be provided.
R.: thank you for the noteworthy opinion. We’ll try to improve the paper according to your remarks.
General concept comments
The section “Material and Methods” should be enhanced information about patients’ cohort. It seems to me, the next data is needed to accurate interpretation of results: patients’ age, size of inguinal hernia and strangulated or not hernia.
R: The section “Material and Methods” has been modified by adding supplementary information as per the reviewer’s suggestion.
The authors use parametric tests such as ANOVA and Tukey for statistical analysis. It is known that these criteria are sensitive to the type of data distribution, while the authors do not assess the normality of the distribution. According to the Materials and Methods section, the statistical analysis is carried out on 9 images for two groups (ST and ET). For such samples, an analysis of the normality of the distribution should be carried out.
R.: According to Materials and Methods section, NGF and NGFR p75 immunohistochemical positive cells were evaluated in 15 biopsies of the 3D dynamic hernia scaffold excised at different times post implantation. Three samples 3-5 weeks ST post-implantation, four samples 3-4 months MT post-implantation, five samples 6-8 months LT post implantation and three ET post-implantation, more than three years after implantation, were analyzed. Three non-overlapping fields were identified for each sample (15). Digital images were then analyzed by Image J computer program to evaluate the percentage of immunohistochemical positive cells to NGF and NGFR p75. Totally we had 15 biopsies, with 3 immunohistochemical images to evaluate, then 45 measures; for this reason, we choose a parametric test and so we assumed the normality of ours measures. Moreover, to detect Gaussian distribution of data, a normality test (Shapiro-Wilk test) was performed. This test resulted positive. These data have been included in the report
It is not clear, what is the mechanism of increasing the nerve growth factor and the neurotrophin receptor?
R.: We are grateful for this interesting observation concerning the mechanism of increasing the nerve growth factor and the neurotrophin receptor. In our study protocol was not planned to carry out "in vitro" or "in vivo" experiments. This decision was motivated by the fact that in literature,several studies have demonstrated that "in vitro" myogenic differentiation, myotube survival and, "in vivo", repair of damaged muscle fibers are activated by cellular signaling pathways NGF / p75NTR axis. In discussion paragraph (page 8 line 283 and following), we have reported the results observed by different research groups which show that NGF exerted a positive effect on myogenesis and NGFR p75 is considered a marker for regenerating fibers in inflamed or dystrophic muscle. Further, we have added two new references in discussion, paragraph concerning these aspects.
Was there a control group of patients who were implanted with a standard mesh?
R.: all patients of the examined cohort were operated with the 3D scaffold ProFlor. No standard meshes were examined.
How is level of the nerve growth factor for these patients? Please, discuss this point.
R.: The biological response of conventional flat meshes has been studied for decades and is unanimously portrayed as a foreign body reaction composed by stiff fibrotic avascular scar plate. Details on the quality of tissue incorporation of conventional meshs are described in the introduction section (L. 75 – 77) where the related literature is also quoted in the reference list. Regarding the level of nerve growth factor in conventional flat meshes, in Introduction section is quoted an article (referenced as #18) that compares the biological response of conventional meshes vs. the 3D scaffold ProFlor. In this report appears evident that only low quality fibrotic incorporation develops in the flat meshes and, above all, there is no evidence of highly specialized tissue structures like muscles and nerves ingrowing in conventional implants.
Specific comments
Figures 2, 3, 4 and 5: Images on these panels have various magnification. Please unify.
R.: I agree with the reviewer, the displayed microphotographs have different levels of magnification. However, we decided to pursue this choice because unifying the magnification of some microphotograph proved inadequate, as the essence of the findings could not be adequately highlighted in a lower degree of magnification. Being confident in the reviewer’s understanding, hope this explanation suffices
Figure 6: Please add name of Y-axes on bar graphs.
R.: done